# Efficient Representations for Whole Slide Image Classification

## Abstract

The advent of digital pathology has transformed diagnostic and research capabilities, offering unprecedented insights through the analysis of high-resolution whole slide images (WSIs). However, the gigapixel size and complexity of WSIs present significant computational challenges. To address this, we propose a scalable and efficient pipeline for WSI classification that integrates patch-based feature extraction, clustering, and advanced representation techniques. Our methodology begins by extracting features from patches identified based on their pathological significance using deep feature embeddings from a pre-trained convolutional neural network (CNN) fine tuned on a histology dataset under noisy labels. This approach ensures that the extracted features are robust and tailored to histopathological patterns despite the inherent noise in the training data. These embeddings are then clustered using K-means clustering to group semantically similar regions. To represent these clusters effectively, we experimented with two strategies: first, using the cluster mean to summarize each cluster; and second, employing Fisher vector (FV) encoding to model the distribution of patch embeddings within clusters using a parametric Gaussian mixture model (GMM). The resulting high-dimensional feature vector encapsulates both local and global tissue structures, enabling robust classification of WSIs. This approach significantly reduces computational overhead while maintaining high accuracy, as validated across multiple datasets. Our innovative framework combines the precision of Fisher vectors with the scalability of clustering, establishing an efficient and precise solution for WSI analysis that advances the practical application of digital pathology in medical diagnostics and research.

## 1 Introduction

Whole Slide Imaging (WSI) has transformed the field of digital pathology by enabling the digitization of histological slides at gigapixel resolution, providing detailed views of entire tissue sections. This technology has facilitated remote diagnostics, educational opportunities, and advanced research, significantly enhancing the visualization, analysis, and management of tissue samples essential for accurate disease diagnosis (Pantanowitz et al., 2011; Malarkey et al., 2015). However, the large size and complexity of WSIs present unique computational challenges, including high memory and storage requirements, as well as difficulties in processing and analyzing the vast amount of data they contain (Brachtel & Yagi, 2012; Kumar et al., 2020). Traditional machine learning and deep learning methods struggle to efficiently process WSIs while preserving their rich structural information, necessitating innovative approaches for feature extraction and classification.

One common solution is to divide WSIs into smaller patches, which are processed independently to reduce computational overhead. While this approach mitigates the challenges posed by WSI size, it risks overlooking broader structural and contextual information crucial for accurate classification. Recent research has focused on integrating patch-level features to capture both local and global tissue characteristics, enabling comprehensive representation of the entire WSI. Building on this foundation, our study introduces a robust framework that explores two complementary methods to address the challenges of WSI analysis.

The first method utilizes cluster mean representation, where WSIs are divided into fixed-size patches (e.g., $512{\times}512$ pixels), and feature embeddings are extracted using pre-trained convolutional neural networks

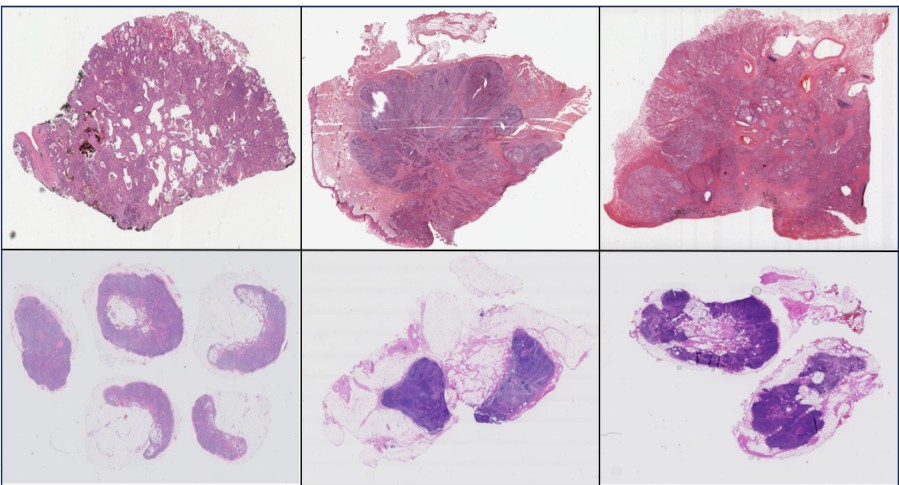

Figure 1: Thumbnail image samples of TCGA Lung data (Coudray et al., 2018) (first row) and Camelyon17 dataset (Bandi et al., 2018a) (bottom row)

(CNNs)(Albawi et al., 2017) or transformers(Khan et al., 2022). These embeddings are grouped using K-means clustering (Lloyd, 1982), where each cluster aggregates semantically similar patches. The cluster means, which summarize the feature distribution within each cluster, are concatenated to form a compact representation of the WSI. This representation is then fed into advanced classifiers, including attention-based multiple instance learning (MIL) (Ilse et al., 2018) and transformer-based models. The self-attention mechanisms in transformers effectively capture spatial relationships among cluster centroids, allowing the classifier to focus on regions most indicative of pathological states. MIL, by treating the WSI as a "bag" of cluster centroids, learns to attend to the most diagnostically relevant clusters, enhancing both interpretability and accuracy.

The second method builds upon the first by replacing the cluster mean representation with a more expressive Fisher vector (FV) encoding. In this approach, patch-level feature embeddings are extracted using a robust feature encoder a pre-trained ResNet34 fine-tuned on a histology dataset. During fine-tuning, patches were assumed to share the same label as their corresponding WSI, with 30% of the labels intentionally treated as noise. This robust feature extractor, trained under noisy label conditions, achieved improved accuracy. In this approach, after patch-level feature embeddings are extracted, K-means clustering is used to group similar patches, as in the first method. However, instead of summarizing each cluster using its mean, Fisher vectors are computed for each cluster using a Gaussian Mixture Model (GMM) (Arun et al., 2020). These FVs capture the distribution of patch embeddings within each cluster, encoding richer local and global information about the WSI. The concatenated Fisher vectors provide a high-dimensional feature vector that serves as input to the same set of classifiers. This substitution significantly enhances the representational power of the model, as Fisher vectors can capture nuanced patterns and relationships within the data, leading to improved classification performance.

Both methods were evaluated on multiple datasets, including TCGA-LUAD (Gupta et al., 2023) for EGFR mutation prediction, and CAMELYON17 (Bandi et al., 2018b) for metastasis detection as shown in Fig 1 and Warwick (Qaiser et al., 2017) for HER2 scoring, TCGA-BRCA (Genomic Data Commons, 2024) for HER2 classification (snapshot of thumbnail is shown Fig 2). The first method demonstrated substantial improvements over traditional patch-based approaches, offering a balance between computational efficiency and classification accuracy. The second method, leveraging Fisher vector encoding, further enhanced these results by providing a richer representation of WSI features. These findings highlight the flexibility and robustness of our framework, paving the way for scalable and insightful WSI analysis, with broad applications in automated disease diagnosis and biomarker discovery. A preliminary version of this work has been reported( (Gupta et al., 2024) for method 1, (Gupta et al., 2025) related to method 2)

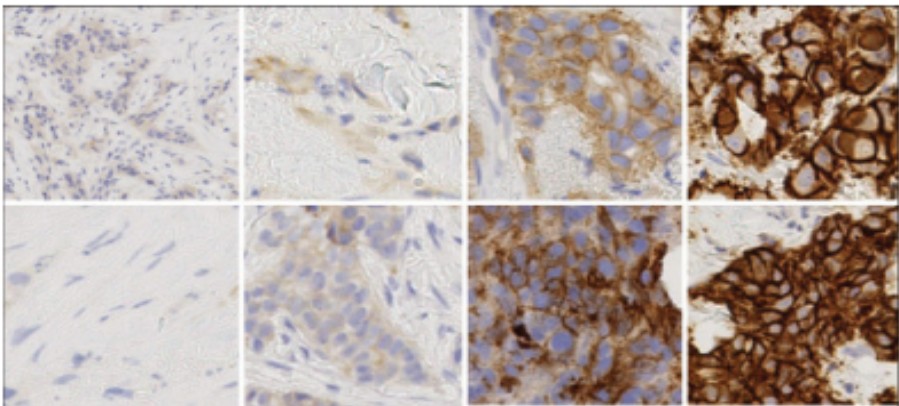

Figure 2: Thumbnail of TCGA HER2neu immunohistochemistry staining, showcasing patches from slides with different HER2 scores, each reflecting varying staining intensities

## 2 Background and Related Work

Extracting robust and discriminative features is crucial for image classification, with approaches ranging from handcrafted to data-learned features. Whole slide image (WSI) classification is often grouped by pooling strategies, underlying assumptions, or specific challenges Dimitriou et al. (2019); Ibrahim et al. (2020).

The immense size of WSIs, often exceeding 100,000 pixels per dimension, necessitates patch-based processing. Under multiple instance learning (MIL), WSIs are represented as "bags" of patches (instances), with various methods assuming patch-level labels align with WSI-level labels. For example,Hou et al. (2016); Combalia & Vilaplana (2018) utilize MIL to classify WSIs by aggregating features from individual patches through CNNs and pooling strategies like max pooling, global average poolingZhou et al. (2016), or k-min/k-max pooling Durand et al. (2016). Pooling strategy significantly impacts performance in both natural Durand et al. (2017) and histopathological images Couture et al. (2018).

Advanced methods like ACMIL Zhang et al. (2024) address limitations in traditional MIL by enhancing discriminative pattern extraction, while PAMT Lin et al. (2024) adapts pre-trained models for histopathology data through prompt-guided transformations. Similarly, PEMP Qu et al. (2024) improves few-shot WSI classification by leveraging visual and textual prompts aligned via vision-language models.

To tackle computational challenges in processing hundreds of thousands of WSI patches, generative models Huang et al. (2019) and efficient dataloaders Akbarnejad et al. (2021) have been proposed. Fisher vector (FV) representations Perronnin et al. (2010) provide robust feature extraction by capturing fine-grained details across multiple regions, addressing the high computational cost of exhaustive patch encoding Lu et al. (2021).

## 3 Proposed Methodology

In this study, we present a comprehensive framework for the classification of whole slide images (WSIs), leveraging two complementary methods to address the challenges of WSI analysis. The methodology encompasses preprocessing, feature extraction, clustering, and classification, with distinct approaches in how cluster representations are used for final classification.

### 3.1 Method 1: Cluster Mean Representation

The first approach focuses on creating a computationally efficient pipeline centered on cluster mean representation for classification shown in Fig 3. Initially, we preprocess WSIs using a deep learning model Patil et al. (2023b) to remove artifacts and non-tissue regions. Subsequently, we apply an automated tissue de-

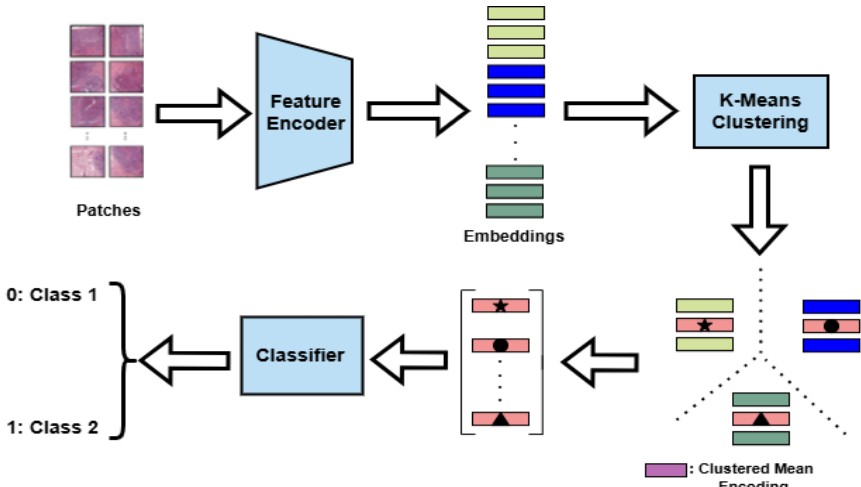

Figure 3: Method1: Patches from WSIs are encoded into embeddings, clustered via K-means. The cluster means are concatenated into a single vector representing the WSI, which is then used for classification.

tection algorithm based on color thresholding and morphological operations using HistomicsTK His. This step retains only diagnostically relevant ROIs.

From the identified ROIs, non-overlapping patches of size $512 \times 512$ pixels are extracted at a $40\times$ zoom level using the OpenSlide library. To enhance patch quality, patches with a nuclei count $< 10$, computed using HistomicsTK, are excluded as they lack significant diagnostic value. This preprocessing step ensures that only high-quality tissue regions are included, filtering out white space and other non-diagnostic areas.

For feature extraction, various pre-trained encoders, including SimCLR Chen et al. (2020), ResNet50 Jian et al. (2016), EfficientNet Tan & Le (2019), RegNet Schneider et al. (2017), ConvNeXT_Tiny Liu et al. (2022), and Swin_Tiny Liu et al. (2021), are used to transform raw pixel data into high-dimensional feature vectors. These vectors are then clustered using the K-means algorithm Lloyd (1982), with the optimal number of clusters determined empirically using the Elbow Method Syakur et al. (2018). In our experiments, we found 10 clusters to be optimal for maintaining a balance between computational efficiency and pathological diversity.

Cluster mean vectors are concatenated to form a comprehensive feature vector for each WSI. To address sensitivity to cluster order, permutation-invariant classifiers like Swin-Tiny Liu et al. (2021), MLP Haykin (1994), and AMIL Ilse et al. (2018) are employed. These enhance robustness and accuracy by focusing on histopathological content and highlighting diagnostically significant WSI regions for pathologists.

### 3.2 Method 2: Fisher Vector Encoding

Method 2 builds on Method 1 by replacing cluster mean representation with Fisher vector (FV) encoding for better feature distribution modeling (Fig.4). Patch-level embeddings were extracted using a fine-tuned ResNet34 on histology data, with 30% noisy labels to improve robustness. After artifact removal and tissue detectionPatil et al. (2023a), WSIs were divided into $512 \times 512$ patches, and embeddings were extracted using pre-trained CNNs and transformer architectures.

After obtaining patch-level feature embeddings, K-means clustering is applied to group patches based on feature similarity, reducing dataset complexity and capturing meaningful tissue patterns. The number of clusters is determined using the Elbow Method, similar to Method 1. For each cluster, Fisher vectors are computed using a Gaussian Mixture Model (GMM) Arun et al. (2020). This process results in a Fisher vector for each feature, capturing the directional rate of change of the log-likelihood. By concatenating these vectors, we obtain a comprehensive and dense representation of the feature distribution. The dimensionality of the resulting Fisher vector is determined by the number of Gaussians in the mixture and the dimensionality

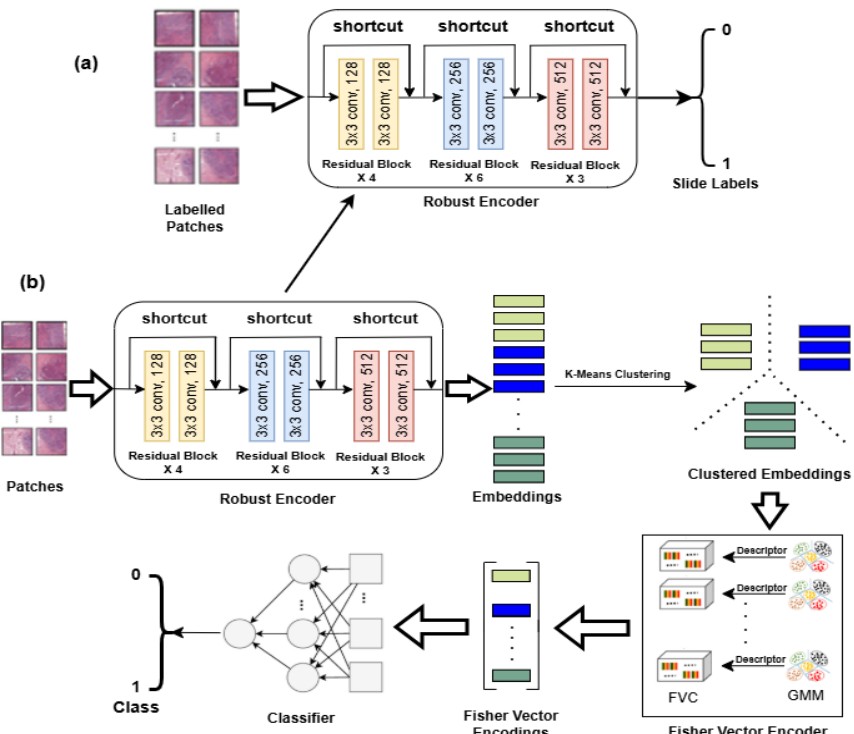

Figure 4: Method2: Patches from WSIs are encoded into embeddings (robust), clustered via K-means, and transformed into Fisher vectors using a GMM model. These Fisher vectors are concatenated into a single vector representing the WSI, which is then used for classification.

of the features (D). Therefore, for a GMM with N Gaussians, the Fisher vector for a D-dimensional feature would have a dimensionality of 2ND (since it includes both mean and covariance information for each Gaussian) Novotnỳ et al. (2015).

In this method, we take into account a predefined set of vectors, denoted as $\{v_1, v_2...., v_M\}$, within the descriptor space. A collection of descriptors is represented by $\{f(x_1), ..., f(x_n)\}$. The Fisher vector is generated by concatenating the normalized gradients of the log-likelihood of local image descriptors concerning the GMM parameters $\mu_m$ and $\sigma_m$ Akbarnejad et al. (2021). Fisher vectors encode the distribution of patch embeddings within a cluster, capturing both the mean and variance of the feature components. The Fisher vector $FV(f(x_i))$ for a set of descriptors $\{f(x_1), \ldots, f(x_n)\}$ is formulated as:

$$FV(.) = \left[ \frac{s_{i1}}{c_1} \left( f(x_i) - v_1 \right), \ldots, \frac{s_{im}}{c_m} \left( f(x_i) - v_m \right), \right.$$
$$\left. \frac{s_{i1}}{\hat{c}_1} \left( f(x_i) - v_1 \right)^2, \ldots, \frac{s_{im}}{\hat{c}_m} \left( f(x_i) - v_m \right)^2 \right] \quad (1)$$

where $c_j$ and $\hat{c}_j$ are scaling constants, and $v_m$ represents GMM coding centers. By modeling the feature space with a GMM, Fisher vectors provide a robust and compact representation of feature distributions in each cluster.

The Fisher vectors from all clusters are concatenated to form a single high-dimensional vector, which aggregates local and global information across the WSI. This vector is then fed into classifiers such as Swin-Tiny Liu et al. (2021), AMIL Ilse et al. (2018), and MLP Haykin (1994), which leverage the rich patch-level and global representations encoded by Fisher vectors. Compared to Method 1, this approach provides a more nuanced understanding of feature distributions, resulting in improved classification accuracy for WSIs.

By combining these two complementary methods, our study demonstrates the flexibility and robustness of feature representation strategies in WSI classification. While Method 1 focuses on computational efficiency and interpretability through cluster means, Method 2 enhances representational power using Fisher vectors, capturing richer local and global patterns. Both methodologies address the challenges of WSI analysis and contribute to scalable, accurate, and interpretable diagnostic pipelines.

## 4 Experimentation and Dataset

### 4.1 Experimentation

We evaluated two methodologies for WSI classification: cluster mean representation and Fisher vector encoding.

Artifacts and non-tissue regions were removed Patil et al. (2023b), and ROIs were identified via HistomicsTK His. Non-overlapping $512 \times 512$ patches were extracted at $40\times$ magnification, excluding patches with nuclei count $< 10$. Features were extracted using encoders like SimCLR Chen et al. (2020), ResNet50 Jian et al. (2016), EfficientNet Tan & Le (2019), RegNet Schneider et al. (2017), ConvNeXT_Tiny Liu et al. (2022), and Swin_Tiny Liu et al. (2021), trained on ImageNet Russakovsky et al. (2015). K-means clustering ($k = 10$) was applied using the elbow method Syakur et al. (2018).

For Method 1, cluster means were concatenated and classified using AMIL, MLP, and Swin-Tiny, optimized with AdamW. Feature augmentation (scaling, jittering, mixup) improved robustness. Evaluations on TCGA-LUAD and CAMELYON17 targeted EGFR mutation and metastasis detection.

Method 2 utilized Fisher vectors from GMM to capture richer distributions, classified with the same models as Method 1. This approach was tested on TCGA-BRCA and Warwick HER2 datasets for HER2 classification and scoring.

### 4.2 Dataset

Our study utilized four diverse datasets, each associated with specific tasks and methodologies:

- **CAMELYON17 (Method 1,2)**: 500 WSIs with four classes (Negative, ITC, Macro-, and Micro-metastases). Binary classification distinguished Metastasis Positive (ITC, Macro, Micro) from Negative cases Litjens et al. (2018).

- **TCGA-LUAD (Method 1,2)**: 159 slides (79 EGFR mutations, 80 Non-EGFR). Binary classification focused on EGFR vs. Non-EGFR mutations Gupta et al. (2023).

- **TCGA-BRCA (Method 2)**: 92 slides (36 HER2-, 56 HER2+). Binary classification distinguished HER2+ from HER2- cases.

- **Warwick HER2 (Method 2)**: 86 WSIs (52 training, 34 testing) from the Warwick HER2 challenge. Tasks included HER2+ vs. HER2- classification and HER2 score prediction.

Across all datasets, rigorous evaluation identified optimal encoders and clustering methods. Feature augmentation and Fisher vector encoding improved accuracy, highlighting the framework's scalability and robustness.

Table 1: Results (%) on the Lung dataset for binary classification with method1. The best performance is marked as bold.

| Feature Extractor | Clustering | Classifier | Accuracy | Precision | Recall |
|---|---|---|---|---|---|
| ResNet-50Jian et al. (2016) | No | MLP | 0.71 | 0.71 | 0.74 |
| ResNet-50 Jian et al. (2016) | Yes | AMIL | **0.83** | 0.81 | 0.83 |
| SimCLR Chen et al. (2020) | Yes | AMIL | 0.81 | 0.80 | **0.84** |
| Swin Tiny Liu et al. (2021) | No | AMIL | 0.75 | 0.75 | 0.75 |
| Swin Tiny Liu et al. (2021) | Yes | AMIL | 0.67 | 0.70 | 0.62 |
| ConvNeXT Liu et al. (2022) | Yes | MLP Haykin (1994) | 0.72 | 0.70 | 0.74 |
| ConvNeXT Liu et al. (2022) | No | AMIL | 0.72 | 0.75 | 0.70 |
| ConvNeXT Liu et al. (2022) | Yes | AMIL | 0.81 | **0.81** | 0.75 |

## 5 Results

The results from our experiments are presented across two methods: cluster mean representation and Fisher vector encoding. Both methods were evaluated on multiple datasets, demonstrating robust classification performance for diverse tasks. A detailed comparison of the outcomes is outlined below.

### 5.1 Method 1: Cluster Mean Representation

Table 1 summarizes the binary classification results for the TCGA-LUAD dataset, focusing on distinguishing EGFR-positive (mutated) and Non-EGFR (wild-type) samples. Using pre-trained backbone feature extractors such as SimCLR Chen et al. (2020), ResNet50 Jian et al. (2016), EfficientNet Tan & Le (2019), RegNet Schneider et al. (2017), ConvNeXT_Tiny Liu et al. (2022), and Swin_Tiny Liu et al. (2021), the pipeline clustered features into 10 clusters using K-means. Each cluster was replaced with its mean vector, which was then fed into classifiers such as Swin-Transformer, MLP, and AMIL.

The proposed method achieved comparable results to the classical approach in Gupta et al. (2023), while significantly reducing memory requirements by representing a gigapixel WSI with a single vector (cluster mean vector). Ablation studies revealed that regions of high cellularity in tumor areas played a critical role in determining the EGFR class, highlighting the utility of clustering for capturing global representations.

Similarly, Table 2 presents the results for the binary classification of metastasis-positive versus metastasis-negative samples in the CAMELYON17 dataset. The same pipeline was applied as for the TCGA-LUAD

Table 2: Results (%) on the Camelyon17 dataset for binary classification with method1. The best performance is marked as bold.

| Feature Extractor | Clustering | Classifier | Accuracy | Kappa Score | Precision | Recall |
|---|---|---|---|---|---|---|
| ResNet-50 Jian et al. (2016) | No | AMIL | 0.72 | 0.54 | 0.73 | 0.72 |
| ResNet-50 Jian et al. (2016) | Yes | AMIL | **0.75** | 0.56 | **0.80** | **0.75** |
| SimCLR Chen et al. (2020) | No | AMIL | 0.64 | 0.48 | 0.64 | 0.66 |
| SimCLR Chen et al. (2020) | Yes | AMIL | 0.68 | 0.50 | 0.70 | 0.68 |
| ConvNeXT Liu et al. (2022) | No | AMIL | 0.58 | 0.42 | 0.60 | 0.60 |
| ConvNeXT Liu et al. (2022) | Yes | AMIL | 0.72 | **0.57** | 0.72 | 0.75 |

dataset. While solid results were obtained, the findings suggest that regions of high cellularity may not always serve as reliable morphological biomarkers for metastasis detection, underlining the distinct challenges posed by metastasis classification tasks.

## 5.2 Method 2: Fisher Vector Encoding

For the Warwick dataset, our model outperformed baselines in HER2 scoring into three classes (negative, equivocal, positive) and binary HER2+ versus HER2- classification tasks, as shown in Tables 3 and 4. The proposed pipeline, combining patch-based feature extraction, K-means clustering, and Fisher vector encoding, demonstrated robustness across both tasks.

Table 3: Results (%) on the Warwick dataset for HER2 score (method2). The best performance is marked as bold.

| Methods | Robust Encoding | Feature Extractor | Accuracy | AUC | Precision | Recall | F1-score |
|---|---|---|---|---|---|---|---|
| DFVC Akbarnejad et al. (2021) | No | ResNet-50 | 0.63 | - | 0.75 | 0.63 | 0.61 |
| AMIL Ilse et al. (2018) | No | ResNet50 | 0.66 | 0.70 | 0.75 | 0.67 | 0.71 |
| Ravi, et al. Gupta et al. (2025) | No | EfficientNetV2-S | 0.72 | 0.71 | 0.77 | 0.75 | 0.76 |
| Proposed method | Yes | EfficientNetV2-S | **0.75** | **0.71** | **0.77** | **0.77** | **0.77** |

Table 4: Results (%) on the Warwick dataset for HER2+ vs HER2- classification (method2). Best performance is in bold.

| Methods | Robust Encoding | Feature Extractor | Accuracy | AUC | Precision | Recall | F1-score |
|---|---|---|---|---|---|---|---|
| Anand, et al. Anand et al. (2020) | No | Neural Network | 0.75 | 0.82 | 0.80 | 0.75 | 0.77 |
| AMIL Ilse et al. (2018) | No | ResNet-50 | 0.78 | 0.80 | 0.75 | 0.81 | 0.80 |
| Ravi, et al. Gupta et al. (2025) | No | RegNetY-3.2GF | 0.80 | 0.83 | 0.91 | 0.90 | 0.90 |
| Proposed Method | Yes | RegNetY-3.2GF | **0.82** | **0.84** | **0.91** | **0.90** | **0.90** |

Similarly, Table 5 highlights the superior performance achieved on the TCGA-BRCA dataset for HER2 status classification. Fisher vector encoding effectively captured local and global feature distributions, resulting in enhanced classification accuracy compared to baseline methods.

To evaluate the adaptability of the approach, experiments were extended to the TCGA-LUAD and CAMELYON17 datasets for EGFR mutation prediction and metastasis detection, respectively. As shown in Table 6, the Fisher vector method achieved strong results, demonstrating its generalizability and applicability in digital pathology.

Table 5: Results (%) on the TCGA-BRCA dataset for HER2+ vs HER2- classification (method2). Best performance is in bold.

| Methods | Robust Encoding | Feature Extractor | Accuracy | AUC | Precision | Recall | F1-score |
|---|---|---|---|---|---|---|---|
| Anand, et al. Anand et al. (2020) | No | ResNet-50 | 0.73 | 0.76 | 0.70 | 0.87 | 0.78 |
| AMIL Ilse et al. (2018) | No | ResNet-50 | 0.76 | 0.64 | 0.67 | 0.75 | 0.71 |
| Sekhar, et al. Sekhar et al. (2024) | No | MoCo-v2 | 0.82 | **0.85** | 0.77 | **0.91** | 0.83 |
| Ravi, et al. Gupta et al. (2025) | No | MoCo-v2 | **0.86** | 0.83 | 0.88 | 0.86 | **0.87** |
| Proposed Method | **Yes** | MoCo-v2 | 0.85 | 0.84 | **0.90** | 0.86 | **0.88** |

Table 6: Results (%) on the different datasets for binary classification (Metastasis detection for CAMELYON17 and Mutation prediction for TCGA-LUAD) with our proposed method2. Best performance is in bold.

| Dataset (Method) | Robust Encoding | Feature Extractor | Accuracy | AUC | Precision | Recall | F1-score |
|---|---|---|---|---|---|---|---|
| TCGA-LUAD (AMIL Ilse et al. (2018)) | No | RegNetY-3.2GF | 0.72 | 0.72 | 0.75 | 0.71 | 0.73 |
| TCGA-LUAD (ISBI Gupta et al. (2025)) | No | RegNetY-3.2GF | 0.84 | 0.79 | 0.75 | 0.80 | 0.77 |
| TCGA-LUAD (Proposed) | Yes | RegNetY-3.2GF | **0.84** | **0.81** | **0.80** | **0.80** | **0.80** |
| CAMELYON17 (AMIL Ilse et al. (2018)) | No | EfficientNetV2-S | 0.75 | 0.70 | 0.73 | 0.73 | 0.73 |
| CAMELYON17 (Ravi, et. al Gupta et al. (2025)) | No | EfficientNetV2-S | **0.77** | 0.71 | 0.73 | 0.73 | 0.73 |
| CAMELYON17 (Proposed) | Yes | EfficientNetV2-S | 0.75 | **0.71** | **0.75** | **0.75** | **0.75** |

Overall, our findings showcase the proposed approaches' flexibility and robustness: the cluster mean method suits computationally efficient tasks, while Fisher vector encoding excels at capturing nuanced features for complex classification.

# 6 Conclusion and Discussion

This study introduces a robust framework for preprocessing, feature extraction, clustering, and classification of WSIs, addressing high dimensionality and limited labeled data. By combining multiple encoders with cluster mean and Fisher vector encoding methods, our approach efficiently captures both local and global tissue features.

The cluster mean representation method offers computational efficiency by summarizing clusters with mean feature vectors, using classifiers like Swin-Tiny, MLP, and AMIL. It balances complexity and accuracy, making it suitable for tasks like EGFR mutation prediction (TCGA-LUAD) and metastasis detection (CAMELYON17), where clustering highlights regions of high cellularity critical for classification.

The Fisher vector encoding method models patch feature distributions within clusters, capturing mean and variance for richer morphological and pathological details. It outperformed the cluster mean approach in HER2 classification and scoring tasks (Warwick, TCGA-BRCA), excelling in heterogeneous datasets but requiring higher computational resources.

Our framework demonstrated strong performance across datasets, with no universally optimal encoder-classifier combination. Transformer-based encoders like Swin-Tiny, coupled with MLP or AMIL, enhanced both accuracy and interpretability, aiding pathologists by focusing attention on diagnostically relevant regions.

Future directions include integrating multimodal data for holistic diagnostics, improving interpretability through visualization tools, and enhancing MIL with hierarchical or spatially aware clustering methods. This flexible and scalable framework has significant potential to automate disease diagnosis, optimize clinical workflows, and personalized medicine.

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
