# OpenReview forum: "Efficient Representations for Whole Slide Image Classification"
_TMLR — Rejected by TMLR_

### Review · Reviewer_vSRn · 2025-04-07

**Summary Of Contributions:**

This paper studies two methods to learn representations from WSIs. The one is cluster mean. The other is Fish vector encoding. Different from most methods for WSI representation learning, the proposed methods mainly rely patch clusters to compute WSI-level representations. Cluster mean directly concatenates the mean embedding of each cluster. Fish vector encoding leverages Fish Vector to describe the features of clusters. Experiments on WSI datasets demonstrate the superiority of the proposed methods over traditional ones.

**Audience:**

No

**Claims And Evidence:**

No

**Requested Changes:**

Please see Weaknesses.

**Strengths And Weaknesses:**

Strengths:
- This paper studies cluster-based representation learning methods for WSIs, which seems interesting.
- Two different methods, cluster mean and Fish Vector encoding, are adopted to describe the clusters.

Weaknesses:
- The writing quality is subpar, which should be improved further. For example, Fig.1 and Fig. 2 seem to present the information that contributes little to the paper; providing the label of WSIs could mitigate this. Moreover, most `\citing` and `\cite` are used incorrectly.
- Experimental settings fail to align with conventional settings. Foundation models (like UNI or CONCH) are generally used for patch feature extraction in the filed of WSI analysis. The CNNs pretrained on limited patches are out-of-date for WSI analysis.
- The second section, Background and Related Work, is unsatisfactory. It should be re-written to cover important literature (e.g., DSMIL, TransMIL, DTFD-MIL, etc) in the field.
- Important baselines for WSI classification are missing, e.g., DSMIL, TransMIL, and DTFD-MIL.
- The author claims that Method 1 focuses on computational efficiency and interpretability through cluster means. However, the experiments fail to demonstrate these.

---

### Review · Reviewer_qK31 · 2025-04-09

**Summary Of Contributions:**

The manuscript proposes two methods for whole slide image classification in histopathology. Both methods extract patch embeddings using pretrained neural networks and compute slide-level representations by first clustering these patch embeddings into 10 clusters using K-Means and subsequently identifying cluster prototypes. The methods differ in the way the prototypes are computed: either by simple averaging or by fitting a Gaussian mixture model. The proposed method could improve the interpretability and efficiency of current approaches, however, none of these aspects are analyzed in the paper. In the current shape, I have major concerns regarding the methodology, the scientific soundness, and the presentation of the results.

**Audience:**

No

**Broader Impact Concerns:**

I have no concerns on the ethical implications of this work.

**Claims And Evidence:**

No

**Requested Changes:**

Major changes:
- As mentioned above, the experimental section should be harmonized across the two methods. Both methods should be evaluated on the same datasets and the same feature extractors.
- Additionally, recent patch-level foundation models should be used as feature extractors as they have been demonstrated to provide better features than ImageNet-pretrained models or supervisedly trained models on small datasets as done in this manuscript.
- Finally, I would expect a qualitative analysis of the 10 (=number of clusters) prototypes and their link to morphological concepts and an ablation study over the number of clusters.

Minor changes:
- The citation style is not optimal as the citations appear in free text without brackets.
- Section 3.2: the formalization of the GMM concept is incomplete: variables f, xi, and si are not introduced, there is a formatting issue in Eq (1), and “,” is missing at the end of the line. Additionally, it would be great to synchronize the method description with the figures. In the figures, the same colors should be used for the same concepts (e.g., Fisher vector encoder).
- Preprocessing descriptions should only be included in the experimental section and should be removed from the method description
- Figure 1 is a duplicate from [3].

**Strengths And Weaknesses:**

Strengths:
- The WSI is efficiently aggregated into prototype vectors summarizing the important morphologies present in the image. Modelling the present morphological concepts with Gaussian mixture models is an interesting approach to building slide-level representations.

Weaknesses:
- While the concept is interesting, it is not novel as for example presented at CVPR 2024 [1], in very early stages at ISBI 2017 [2], or at BIBE 2024 [3].
- Related work: The current state of the field is not adequately pictured in the related work section: Patch-level foundation models [4,5,…], methods for slide-level representation learning either self-supervised or unsupervised [1,6,7…], and recent multiple instance learning methods are not mentioned.
- Methodology:
  - Similarly, the proposed unsupervised clustering methods are not evaluated with embeddings of state-of-the-art pathology models such that it remains unclear if the method improves.
  - Simple unsupervised baselines such as mean pooling or more advanced unsupervised prototyping methods [1] are not mentioned or compared to.
- Scientific soundness: Overall, there are several choices in design and evaluation that are unclear to me:
  - Why are different feature extractors used for the two proposed clustering methods?
  - There are no details on the choice of the feature extractors or the pretraining with noisy labels (except for the architecture: ResNet34) that is performed for Method 2, such as the pretraining dataset used.
  - Why are the different clustering methods evaluated on different datasets and not in the same setting, such that a comparison is possible?
  -  There is no standard deviation given across different splits or multiple runs such that the stochasticity of the evaluations can be assessed.
  - There are ablation studies described but not shown or referenced in the paper.
  -  How were the datasets selected and the labels curated? E.g., the cohort TCGA LUAD contains more slides (and patients) with labels for EGFR than the 159 slides mentioned. I assume the cases were selected such that the dataset is balanced but this selection process is not described.
  - Is the TCGA-BRCA HER2 task based on immunohistochemistry WSIs as shown in Fig 2? This is not mentioned in the dataset description or elsewhere. Generally, there is no information on the staining. While this is acceptable if all tasks are done on the commonly used routine-H&E staining, this is not acceptable if parts of the datasets diverge from this assumption.

References:

[1] Song, Andrew H., et al, “Morphological Prototyping for Unsupervised Slide Representation Learning in Computational Pathology”, CVPR 2024

[2] Song, Yang, et al. "Adapting fisher vectors for histopathology image classification." 2017 IEEE 14th international symposium on biomedical imaging (ISBI 2017). IEEE, 2017.

[3] Gupta, Ravi Kant, et al. "Efficient Whole Slide Image Classification Through Fisher Vector Representation." 2024 IEEE 24th International Conference on Bioinformatics and Bioengineering (BIBE). IEEE, 2024.

[4] Chen, Richard J., et al. "Towards a general-purpose foundation model for computational pathology." Nature Medicine 30.3 (2024): 850-862.

[5] Vorontsov, Eugene, et al. "A foundation model for clinical-grade computational pathology and rare cancers detection." Nature medicine 30.10 (2024): 2924-2935.

[6] Xu, Hanwen, et al. "A whole-slide foundation model for digital pathology from real-world data." Nature 630.8015 (2024): 181-188.

[7] Wang, Xiyue, et al. "A pathology foundation model for cancer diagnosis and prognosis prediction." Nature 634.8035 (2024): 970-978.

---

### Review · Reviewer_AF3S · 2025-04-18

**Summary Of Contributions:**

The paper proposes a scalable pipeline for classifying gigapixel whole slide images (WSIs) by extracting patch‐level embeddings using pre‐trained CNNs and vision transformers, clustering these embeddings via K‑means, and forming slide‑level representations either by concatenating cluster means or by encoding richer first‑ and second‑order statistics with Gaussian‑mixture Fisher vectors. Evaluated on TCGA‑LUAD (EGFR mutation), CAMELYON17 (metastasis), TCGA‑BRCA and Warwick HER2 (HER2 status and scoring), both methods markedly reduce computational burden while maintaining competitive accuracy, with the Fisher vector approach yielding the highest overall performance across diverse pathology tasks.

**Audience:**

Yes

**Claims And Evidence:**

No

**Requested Changes:**

* Based on the above comments, I think significant effort is required to improve the empirical evaluations by expanding the set of comparative baselines and reporting measures of variability. Specifically, the authors should benchmark their methods against recent, state‑of‑the‑art MIL frameworks—such as Attention‑based Deep MIL (ABMIL), Dual‑stream MIL (DSMIL), and Transformer‑based MIL (TransMIL)—to demonstrate genuine advances over established approaches. Moreover, given the small sample sizes and known run‑to‑run fluctuations in WSI studies, it is critical to report standard deviations or confidence intervals across multiple random seeds or cross‑validation folds (e.g., mean ± SD) so that readers can assess the statistical robustness of the results. Incorporating these additional baselines and variability metrics will substantially bolster the study’s rigor, credibility, and reproducibility.
* The authors need a siginicant efforts to highlight their methodological contribution and disguish them from existing work. The current workflow, from my opinion, is just a combination of standard MIL pipeline for WSI analysis.

**Strengths And Weaknesses:**

**Strengths**

* Tackling pathology data with multiple instance learning is clinically important

**Weaknesses**

* The manuscript offers little in the way of methodological innovation, instead recycling well‑established techniques—relying solely on conventional CNN backbones (e.g., EfficientNet) and the ubiquitous combination of K‑means clustering with GMM‑based Fisher vector aggregation—while failing to incorporate more recent transformer‑based architectures that have become standard in cutting‑edge whole‑slide image analysis.
* The empirical validation is insufficiently rigorous, as the authors do not benchmark their methods against current state‑of‑the‑art MIL baselines—namely Attention‑based Deep MIL (ABMIL[1]), Dual‑stream MIL (DSMIL), and Transformer‑based MIL (TransMIL)—making it impossible to contextualize any claimed performance gains. Furthermore, the results omit any measure of variability (e.g., standard deviations over multiple runs or cross‑validation folds), despite the small sample sizes and well‑documented run‑to‑run fluctuations in WSI tasks.

[1] Ilse, Maximilian, Jakub Tomczak, and Max Welling. "Attention-based deep multiple instance learning." International conference on machine learning. PMLR, 2018.
[2] Shao, Zhuchen, et al. "Transmil: Transformer based correlated multiple instance learning for whole slide image classification." Advances in neural information processing systems 34 (2021): 2136-2147.
[3] Li, Bin, Yin Li, and Kevin W. Eliceiri. "Dual-stream multiple instance learning network for whole slide image classification with self-supervised contrastive learning." Proceedings of the IEEE/CVF conference on computer vision and pattern recognition. 2021.

---

### Decision · Action_Editor_MHJT · 2025-05-27

**Recommendation:** Reject

**Comment:**

The reviewers raised various valid concerns, particularly regarding methodological novelty and insufficient empirical validation/comparisons. The authors did not respond during the discussion period, missing an opportunity to clarify or address these points. While the topic may interest the TMLR audience, the work requires significant revision.

Specifically, the method’s unique contributions should be better motivated and validated through better comparisons and more rigorous experiments (e.g., multiple seeds). Improvements in clarity and presentation would also strengthen a future submission.

**Audience:**

I agree with the reviewers that the paper would be overall suited for the TMLR audience.

**Claims And Evidence:**

The reviewers agree that the paper does not meet TMLR’s standards with respect to Claims And Evidence. For instance, various relevant baseline comparisons were proposed by the reviewers, but not followed up on by the authors during the discussion phase. This could include “naiive” baselines like mean pooling as well as relevant published work mentioned by the reviewers. In addition, the reviewers remarked that additional experiments should be conducted to judge the variability of the results (e.g., across model seeds), as well as revisit the cross-validation scheme. Finally, some comments about methodological advances should have been addressed through an update in the manuscript, and additional comparisions and discussions of related work.

Overall, I agree that the key claim in the abstract that the "approach significantly reduces computational overhead while maintaining high accuracy, as validated across multiple datasets" does not hold up within the current experimental setup.

**Resubmission Of Major Revision:**

The authors may consider submitting a major revision at a later time.